# Comparative Genomics Reveals Evidence of the Genome Reduction and Metabolic Potentials of *Aliineobacillus hadale* Isolated from Challenger Deep Sediment of the Mariana Trench

**DOI:** 10.3390/microorganisms13010132

**Published:** 2025-01-10

**Authors:** Shaofeng Yang, Jie Liu, Yang Liu, Weichao Wu, Jiahua Wang, Yuli Wei

**Affiliations:** 1College of Oceanography and Ecological Science, Shanghai Ocean University, Shanghai 201306, China; 2Marine Biomedical Science and Technology Innovation Platform of Lingang Special Area, Shanghai 201306, China; y_liu@shou.edu.cn

**Keywords:** genome streamlining, *Bacillota*, genomic comparison, Mariana Trench

## Abstract

Hadal zones account for the deepest 45% of oceanic depth range and play an important role in ocean biogeochemical cycles. As the least-explored aquatic habitat on earth, further investigation is still required to fully elucidate the microbial taxonomy, ecological significance, metabolic diversity, and adaptation in hadal environments. In this study, a novel strain Lsc_1132^T^ was isolated from sediment of the Mariana Trench at 10,954 m in depth. Strain Lsc_1132^T^ contains heterogenous 16S rRNA genes, exhibiting the highest sequence similarities to the type strains of *Neobacillus drentensis* LMG 21831^T^, *Neobacillus dielmonensis*, *Neobacillus drentensis* NBRC 102427^T^, *Neobacillus rhizosphaerae*, and *Neobacillus soli* NBRC 102451^T^, with a range of 98.60–99.10% identity. The highest average nucleotide identity (ANI), the highest digital DNA-DNA hybridization (DDH) values, and the average amino acid identity (AAI) with *Neobacillus* sp. PS3-40 reached 73.5%, 21.4%, and 75.54%, respectively. The major cellular fatty acids of strain Lsc_1132^T^ included iso-C_15:0_, Summed Feature 3 (C_16:1_*ω*6c and/or C_16:1_*ω7*c), iso-C_17:0_, anteiso-C_15:0_, and iso-C_17:1_*ω*5c. The respiratory quinone of strains Lsc_1132^T^ was MK-7. The G + C content of the genomic DNA was 40.9%. Based on the GTDB taxonomy and phenotypic data, strain Lsc_1132^T^ could represent a novel species of a novel genus, proposed as *Aliineobacillus hadale* gen. nov. sp. nov. (type strain Lsc_1132^T^ = MCCC 1K09620^T^). Metabolically, strain Lsc_1132^T^ demonstrates a robust carbohydrate metabolism with many strain-specific sugar transporters. It also has a remarkable capacity for metabolizing amino acids and carboxylic acids. Genomic analysis reveals a streamlined genome in the organism, characterized by a significant loss of orthologous genes, including those involved in cytochrome c synthesis, aromatic compound degradation, and polyhydroxybutyrate (PHB) synthesis, which suggests its adaptation to low oxygen levels and oligotrophic conditions through alternative metabolic pathways. In addition, the reduced number of paralogous genes in strain Lsc_1132^T^, together with its high protein-coding gene density, may further contribute to streamlining its genome and enhancing its genomic efficiency. This research expands our knowledge of hadal microorganisms and their metabolic strategies for surviving in extreme deep-sea environments.

## 1. Introduction

The hadal zone, located at water depths below 6000 m, accounts for the deepest 45% of the ocean’s depth range and hosts active and diverse biological communities, thereby holding significant importance in the ocean ecosystem [1]. This geological feature, particularly the Challenger Deep, is distinguished by its extreme environmental parameters, including pressures that surpass 110 MPa, perpetual darkness, and temperatures that hover just above the eutectic point of seawater [2,3]. Despite these conditions, the hadal sediments of the Mariana Trench harbor a complex and, to date, poorly characterized assemblage of microorganisms [4,5]. Here, microbiota have adapted to the oligotrophic conditions through a suite of trophic strategies that are essential for survival amidst nutrient-depleted surroundings [6]. The hadal heterotrophic microbes proliferate by decomposing the sparse particulate organic matter (POM) that sinks from the euphotic and disphotic zones, a process integral to nutrient recycling within this impoverished ecosystem [7,8]. Moreover, some microbes exhibit the capacity to metabolize refractory organic compounds, further enhancing nutrient availability. Simultaneously, a distinct group of hadal microorganisms has evolved chemosynthetic pathways, tapping into the thermodynamic gradients of inorganic compounds like hydrogen sulfide. The metabolic activities of microorganisms in the dark ocean depend on the availability and speciation of electron donors (oxidizable compounds) and acceptors (reducible compounds) [9]. Electrochemical investigations have highlighted that nitrate is the favored electron acceptor for chemolithoautotrophic microorganisms, with alternative options including manganese(IV), iodate, iron(III), and sulfate [10]. In environments characterized by low oxygen concentrations, these microorganisms flourish by utilizing oxygen ions derived from the reduction in high-valence salts, specifically nitrate [11] and sulfate [12], thereby serving as electron acceptors in their metabolic processes. These microorganisms are instrumental in the hadal ecosystem, as they catalyze energy-generating reactions independent of sunlight, laying the foundation for a specialized food web that is uniquely tailored to the oligotrophic deep-sea environment [13,14].

The *Bacillota*, also known as phylum *Firmicutes* [15], is a highly diverse group of Gram-stain-positive bacteria that are characterized by their ability to form endospores, which are highly resistant and dormant structures, allowing for these bacteria to survive in extreme conditions, such as high temperatures, desiccation, radiation, and chemical stress. Currently, *Bacillota* contains seven classes, including *Bacilli*, *Clostridia*, *Culicoidibacteria*, *Limnochordia*, *Negativicutes*, *Thermolithobacteria*, and *Erysipelotrichia* [16,17]. The ecological roles of *Bacillota* in the hadal ecosystem are multifaceted. They contribute to the biogeochemical cycling of elements such as nitrogen and sulfur, with some species capable of ammonia oxidation and other autotrophic metabolic processes, suggesting that microbial carbon fixation may be a significant source of organic carbon in the hadal sediment. Despite these insights, further investigation is still required to fully elucidate the taxonomy, ecological significance, metabolic diversity, and adaptation of *Bacillota* species, especially in the hadal zone, which is one of the most extreme environments on Earth [18].

*Neobacillus*, a novel genera belonging to the family *Bacillaceae*, was proposed in 2020. The type species of *Neobacillus* is *N. niacin*, which was isolated from soil in 1991 and previously classified within genus *Bacillus* [19]. Subsequently, *N. novalis* and *N. vireti* were isolated from agricultural soil [20], *N. citreus* and *N. rhizophilus* were isolated from citrus rhizosphere soil [21]. Recently, *Bacillus piezotolerans* YLB-04^T^ were also reclassified as *Neobacillus* species after taxonomic revisions [22]. This strain was originally isolated from the hadal zone, which lies at water depths below 6000 m and was considered to be a piezotolerant bacterium [23].

Currently, the genus *Neobacillus* contained 25 species with validated names (https://lpsn.dsmz.de/search?word=Neobacillus) (accessed on 7 January 2025). However, due to limited research, many non-type strains currently classified within the *Neobacillus* genus show significant differences from the type strains. For instance, the highest ANI value between *Neobacillus* sp. PS3-12 and other type strains is only 71.35%, which is lower than the ANI values between type strains. This suggests that the phylogenetic status of the strains within the *Neobacillus* genus may need to be reclassified.

In this research, we report a novel bacterial strain within *Bacillota*, designated as strain Lsc_1132^T^, isolated from the sediments within the Mariana Trench with a depth of 10,954 m. Phylogenetic analysis has revealed that it could represent a new genus that is highly related to, but not included in genus *Neobacillus*. We further undertook an exhaustive analysis to elucidate its genomic attributes and metabolic capabilities. It is noteworthy that its genome exhibits a profound reduction in size compared to its closest phylogenetic neighbors, which led us to investigate the genetic underpinnings of its evolutionary history and the adaptive strategies it employs to thrive in the hadal environment. This study offers valuable insights into the genomic and metabolic characteristics of microorganisms residing in the deep biosphere, and enhances our comprehensive understanding of how hadal microbes adapt to extreme environments.

## 2. Materials and Methods

### 2.1. Sample Description

Sediment samples were collected at a depth of 10,954 m (11.327° N, 142.188° E) (Figure 1) in the Mariana Trench during the cruise aboard R/V “TAN SUO YI HAO” in November 2019. To collect sediment from the seafloor, after the lander reached the seafloor, the box corer was slowly driven into the seafloor until it reached about 25 cm below the sediment surface. A lid was then released to seal the box corer and the lander was recovered. Collected sediment was immediately subsampled onboard at 2–10 cm depth intervals for geochemical and microbiology analyses.

### 2.2. Bacterial Isolation

The samples were diluted at a rate of 1:100 and incorporated into Marine Broth 2216E culture medium supplied with additional carbohydrates (including 0.05 g/L D-raffinose, 0.05 g/L D-mannitol, and 0.05 g/L D-mannose). After a three-week incubation period at room temperature, the microbial enrichments were diluted further to a ratio of 1:1000 using ASW. They were then inoculated onto MB 2216E-agar plates, which were composed of (1000 mL seawater, 5 g Peptone, 1 g Yeast Extract, and 15 g Agar, with a pH of 7.5–7.6). The inoculated plates were incubated at 30 °C for two weeks. Subsequently, colonies were isolated and purified through the method of streaking. The particular strain designated as strain Lsc_1132^T^ was routinely cultivated on MB 2216E medium under aerobic conditions.

The artificial seawater (ASW) used in this study contained 52 g/L NaCl, 15 g/L Agar, 10 g/L MgCl_2_·6H_2_O, 8 g/L Na_2_SO_4_, 1 g/L KCl, 2.8 g/L CaCl_2_·2H_2_O, 0.6 g/L NH_4_Cl, 0.2 g/L KH_2_PO_4_, and 2 mM NaHCO_3_, which were autoclaved and supplemented with mixed vitamins and trace elements (1:1000) sterilized using 0.2 μm filter. Finally, the strain Lsc_1132^T^ was stored at −80 °C in liquid medium supplemented with 20% (*v*/*v*) glycerol.

To examine the growth trajectories of strain Lsc_1132^T^, we cultured it in MB 2216E medium for approximately 24 h at 37 °C to achieve the logarithmic growth phase. We introduced a small volume of the culture broth into the medium designated for growth curve analysis and adjusted the initial optical density (OD) to approximately 0.05. The growth profiles at various temperatures were recorded using MB 2216E medium (pH = 7.5), with the temperatures set at 4, 20, 37, and 45 °C. The pH-dependent growth curves were established by modifying the pH of MB 2216E medium to 2–10 using HCl or NaOH, while maintaining the incubation temperature at 37 °C. To establish the NaCl concentration range essential for the growth of strain Lsc_1132^T^, we initially formulated a MB 2216E medium devoid of NaCl. Subsequently, we introduced a precise quantity of NaCl to yield final concentrations from 0% to 23%. The incubation temperature was maintained at 37 °C and the pH was adjusted to 7.5. Each of the above experiments was conducted with three biological replicates, and the measurements of OD_600_ represent the average of three readings for each sample.

### 2.3. Microbial Utilization of the Substrates and Chemotaxonomy

Strain Lsc_1132^T^ was also characterized using 94 different biochemical tests on Biolog GEN III MicroPlate (Biolog Inc., Hayward, CA, USA). The phenotypic profiles generated using the visible purple wells were compared with the data in the Biolog database (version 2.8).

Moreover, the capacity of strain Lsc_1132^T^ to metabolize arabinogalactan was tested via challenging it with 1.0 g/L arabinogalactan as the sole carbon source. In detail, log-phase culture of strain Lsc_1132^T^ was washed with ASW three times and then was aliquoted into arabinogalactan-containing medium for cultivation at 37 °C. In total, 50 µL of the bacterial culture broth at day 0 (before cultivation) and day 5 (after cultivation) was coated onto LB agar plates for incubation at 37 °C. Finally, the colony numbers of strain Lsc_1132^T^ were counted on these plates. Each sample was tested with three independent biological replicates.

The extraction and analysis of respiratory quinones were conducted using the methodology outlined by Minnikin et al. [24] and the high-performance liquid chromatography (HPLC) procedure described by Tindall [25].

In the context of cellular fatty acid profiling, strain Lsc_1132^T^ was cultured in LB medium for 48 h at 37 °C. The saponification, methylation, and extraction of fatty acids were carried out according to the standard operating procedure of the MIDI (Sherlock Microbial Identification System, version 6.0). Gas chromatography was performed using an Agilent Technologies 6850 instrument (Agilent, Santa Clara, CA, USA)to analyze the fatty acid methyl esters, and the identification of these compounds was facilitated by the RTSBA6.0 database, which is part of the Microbial Identification System (Athalye et al., 1984) [26].

### 2.4. Genomic DNA Extraction, Sequencing, and Assembly

The genomic DNA of strain Lsc_1132^T^ was extracted following the protocol described by Fang et al. [27]. The genome sequencing of strain Lsc_1132^T^ was carried out by MajorBio (Shanghai Majorbio Bio-pharm Technology Co., Ltd., Shanghai, China) utilizing both the PacBio RS II Single Molecule Real-Time (SMRT) technology from Pacific Biosciences in Menlo Park, CA, USA, and the Illumina HiSeq 2500 platform provided by Illumina Inc., located in San Diego, CA, USA.

The finished genome adopts a hybrid sequencing approach combining second-generation (Illumina) and third-generation (PacBio) technologies. For each sample, no less than 100× coverage of PacBio sequencing data and 100× coverage of Illumina sequencing data are provided simultaneously. This ensures a more complete and accurate assembly of the genome. The hybrid approach allows for the finished genome to avoid the loss of information from small plasmids (<15 kb), thereby guaranteeing the acquisition of a complete genome that includes plasmid sequences.

For Illumina sequencing, Illumina sequencing libraries were prepared from approximately 1 μg genomic DNA sheared into 400–500 bp fragments using a Covaris M220 Focused Acoustic Shearer, following the manufacturer’s protocol. The NEXTFLEXTM Rapid DNA-Seq Kit (NEXTFLEX, San Jose, CA, USA) was used for library preparation.

For Pacific Biosciences sequencing, an aliquot of 15 μg DNA was spun in a Covaris g-TUBE (Covaris, Woburn, MA, USA) at 6000 RPM for 60 s using an Eppendorf 5424 centrifuge (Eppendorf, New York, NY, USA). The DNA fragments were then purified, end-repaired, and ligated with SMRTbell sequencing adapters following the manufacturer’s recommendations (Pacific Biosciences, CA). The sequencing library was purified three times using 0.45 × volumes of Agencourt AMPure XP beads (Beckman Coulter Genomics, Danvers, MA, USA), again following the manufacturer’s recommendations. A ~10 kb insert library was prepared and sequenced on one SMRT cell using standard methods.

The data generated from both PacBio (221,897 reads) and Illumina platforms (4,773,982 pairs of reads with 150 nt) were utilized for bioinformatics analysis. The raw Illumina sequencing reads generated from the paired-end library were subjected to quality-filtered using fastp v0.23.0. The HiFi reads were generated from the PacBio platform for analysis. Then, the clean short-reads and HiFi reads were assembled to construct complete genomes using Unicycle v0.4.8 [28] and Pilon v1.22 to polish the assembly using short-read alignments, reducing the rate of small errors. The final assembled genome was submitted to the NCBI database (version 264.0) (accession number CP172536.1). Genome coverage is 91.89%. The coverage of this genome was calculated using Bowtie 2 [29] to align the reads to its chromosome. Then, Samtools [30] was utilized to convert SAM files into BAM format, sort the BAM files (https://github.com/samtools/hts-specs) (accessed on 7 January 2025), and establish their indexes. The BamM was employed to filter through the aligned reads, retaining only those with a coverage of at least 90% and an identity of at least 95%. The average coverage per base pair of the contigs was determined by “parse” subcommand of BamM, which facilitated the exclusion of the highest and lowest 10% coverage areas using the “tpmean” parameter.

### 2.5. 16S rRNA Prediction and Comparison

The 16S rRNA sequences of strains Lsc_1132^T^, EB600, PS3-12, and PS3-40, as well as 20 *Neobacillus* type strains, were predicted from their genomes using Barrnap (https://github.com/tseemann/barrnap) (accessed on 7 January 2025). The sequences with less than 1500 nt were dropped. The 16S rRNA similarities were calculated using BLASTn [31]. Considering that some of these genomes contain heterogeneous 16S rRNA sequences, only the max values of identities between two strains were shown.

### 2.6. Gene Annotation and Genomic Comparison

The NCBI prokaryotic genome annotation pipeline [32] was employed for ORF prediction and gene annotation. Additionally, the predicted protein sequences were aligned with the Clusters of Orthologous Groups of proteins (COG) [33] and TransporterDB 2.0 [34] databases using BLASTp software (version 2.9.0), with parameters set for identity (50%), query coverage (50%), e-value (1e-5), and score (40) [35]. The Kyoto Encyclopedia of Genes and Genomes (KEGG) annotation was assigned using Blast KOALA [36]. Genomic islands were predicted with Island Viewer 4 [37].

Using a local OrthoMCL 2.0.9 [38], protein families of strain Lsc_1132^T^ and its phylogenetically related strains were clustered with specific cutoff values: identity at 50%, query coverage at 50%, e-value at 1e-10, score at 40, and MCL inflation at 1.5. Protein families that were found to be employed by only one strain were designated as strain-specific. Average nucleotide identity (ANI) was calculated using JSpeciesWS [39], and average amino acid identity (AAI) was calculated using CompareM (https://github.com/dparks1134/comparem) (accessed on 7 January 2025). The genome-to-genome distance (DDH) was calculated with GGDC 3.0 [40]. The whole-genome sequence collinearity analysis of the strains was conducted using Mauve (version 2.3.1) [41].

### 2.7. Phylogenetic Analysis

To elucidate the phylogenetic relationships of strain Lsc_1132^T^ and its associated strains, we used a dataset of 120 conserved bacterial marker genes as defined by the Genome Taxonomy Database (GTDB). We identified the genomic sequences encoding these 120 marker proteins using the GTDB-Tk toolkit with the reference database version Release 07-RS207 [42]. These sequences were then individually aligned using Clustal Omega [43]. After alignment, we eliminated positions with gap percentages of 50% or higher with trimAL [44] and compensated for any missing marker proteins in certain genomes by inserting an equivalent number of “-” to align the sequence lengths after gap removal. The alignments, now gap-free, were concatenated for each marker protein. The concatenated alignment was used to construct a phylogenetic tree using the neighbor-joining algorithm implemented in FastTree2 [45]. To assess the robustness of the tree, a bootstrap test with 1000 iterations was performed. Finally, the phylogenetic tree was visualized using the Interactive Tree of Life (iTOL v5) software [46].

### 2.8. Gene Gain and Loss Analysis

The protein families of strains Lsc_1132^T^, EB600, PS3-12, and PS3-40, as well as 20 *Neobacillus* species, were clustered using OrthoMCL 2.0.9 with the following parameters: identity 50%, coverage 70%, e-value 1e-10, score 40, and inflation index 1.4. A subtree including these 24 strains was extracted from the full phylogenetic tree and saved as a Newick file using MEGA 11 [47]. The ancestral reconstruction and gene content analysis for the evolutionary tree these strains were calculated using Count [48] based on Dollo parsimony.

## 3. Results and Discussion

### 3.1. Description of Strain Lsc_1132^T^

Cells of strain Lsc_1132^T^ were Gram-stain-positive, non-motile, lacking flagella, rod-shaped, non-pigmented, 0.8–1.2 μm in width, and 2–4 μm in length (Appendix A). Strain Lsc_1132^T^ produced a colony that was white circular and featured regular slightly elevated edges. Cells grow in the presence of 0–3% (*w*/*v*) (optimum, 0%) NaCl, at 15–45 °C (optimum, 40 °C), and pH 5–8 (optimum, pH 6) (Figure 2). Differentiating phenotypic characteristics of strain Lsc_1132^T^ from its closest phylogenetic neighbors are shown in Table 1.

When assayed with the GEN III MicroPlate kit, positive for utilization of pectin, dextrin, D-maltose, D-trehalose, D-cellobiose, sucrose, D-turanose, stachyose, D-raffinose, *α*-D-lactose, D-melibiose, *β*-methyl-D-glucoside, D-salicin, *N*-acetyl-D-glucosamine, *N*-acetyl-*β*-D-mannosamine, *N*-acetyl-D-galactosamine, *N*-acetyl neuraminic acid, *β*-gentiobiose, *α*-D-glucose, D-mannose, D-fructose, D-galactose, 3-methyl glucose, D-fucose, L-fucose, L-rhamnose, and D-mannitol (Appendix A). The major cellular fatty acids of strain Lsc_1132^T^ (>5.0%) included iso-C_15:0_ (41.8%), Summed Feature 3 (C_16:1_*ω*6c and/or C_16:1_*ω*7c; 8.2%), iso-C_17:0_ (6.4%), anteiso-C_15:0_ (5.5%), and iso-C_17:1_*ω*5c (5.5%); (Appendix A). The respiratory quinone of strains Lsc_1132^T^ was MK-7, which was similar to the type of strains within genus *Neobacillus* [49].

### 3.2. The Phylogenetic Characteristics of Strain Lsc_1132^T^

Despite multiple attempts at purifying the colony of strain Lsc_1132^T^, when we conducted PCR amplification of the 16S rRNA gene using degenerate primers, the sequencing chromatograms consistently displayed polyclonal patterns. This suggests that there could be a presence of intra-strain heterogeneity in the 16S rRNA gene sequences. Therefore, we employed a T-A cloning strategy to obtain discrete amplicon clones, and a nearly complete 16S rRNA gene sequence (1367 nt) was obtained. The predicted full-length rRNA sequences of strain Lsc_1132^T^ were found to have highest sequence similarity to *Neobacillus bataviensis* LMG 21833^T^ (AJ542508) (99.10%), *N. dielmonensis* (HG315676) (99.02%), and *N. drentensis* NBRC 102427^T^ (AJ542506) (99.01%) (Appendix A). In addition, it exhibited both the highest average nucleotide identity (ANI), the highest digital DNA-DNA hybridization (DDH) values, and average amino acid identity (AAI) with *Neobacillus* sp. PS3-40, reaching 73.5%, 21.4%, and 75.54%, respectively (Appendix A), which are far below the cutoff values recommended for bacterial species delineation. The phylogentic tree, based on the 16S rRNA, is shown in Appendix A.

Moreover, the heterogeneity of its 16S rRNA genes might affect the determination of its phylogenetic position. Therefore, we performed whole-genome sequencing on strain Lsc_1132^T^ and employed the Genome Taxonomy Database (GTDB) classification method to identify its phylogeny. This indicated that strain Lsc_1132^T^, two misclassified strains, “*Neobacillus* sp. PS3-12” isolated from a rice paddy field and “*Bacillus* sp. EB600” from compost, as well as two metagenome-assembled genomes (MAGs), were clustered into a distinct phylogenetic branch. In GTDB taxonomy, this branch was closely related to, but not included in, *Neobacillus*, which currently lacks a taxonomically valid name and is designated as genus “*JAEVLT01*” within the family “*DSM-18226*” of the order “*Bacillales_B*” according to the GTDB taxonomy. The genome of strain Lsc_1132, with a compact size of 3,735,591 base pairs (bp), is notably smaller than those of strains EB600 and PS3-12 (5,828,786 and 5,612,023 bp, respectively), as well as the outgroup strain PS3-40 (4,459,106 bp) within genus “*JAEVLT01*” that also lacks a taxonomically valid name and is closely related to strain Lsc_1132 (Figure 3).

Moreover, strain Lsc_1132^T^ differs significantly from *Neobacillus* species in terms of fatty acid composition. For instance, anteiso-C_15:0_ and C_16:1_*ω*7c alcohol account for only 5.5% and 0.3%, respectively, in strain Lsc_1132^T^, but are at least 12.5% and 2.3% in the five type strains of genus *Neobacillus*. On the contrary, iso-C_17:0_ accounts for 6.4% in strain Lsc_1132^T^, while it does not exceed 2.7% in *Neobacillus* species. Furthermore, iso-C_17:1_*ω*5c is present at 5.5% in strain Lsc_1132^T^, but it was undetectable in *Neobacillus* species (Table 1). These pieces of evidence collectively indicate that strain Lsc_1132^T^ could represent a novel species of a novel genus, proposed as *Aliineobacillus hadale* gen. nov. sp. nov. (type strain Lsc_1132^T^ = MCCC 1K09620^T^).

### 3.3. Description of Genomic Features

The genome of strain Lsc_1132^T^, with a compact size of 3,735,591 base pairs (bp), is notably smaller than those of strains EB600 and PS3-12 (5,828,786 and 5,612,023 bp, respectively), as well as the outgroup strain PS3-40 (4,459,106 bp) that was within the genus “*JAUZPL01*” and closely related to strain Lsc_1132^T^ (Figure 3). Moreover, genomic collinearity analysis indicates that the genomic homology between Lsc_1132^T^ and both PS3-40 and PS3-12 is notably reduced, accompanied by pronounced gene rearrangement events (Figure 4). The G + C content of strain Lsc_1132^T^ genome is 40.9%, which is slightly higher than strains EB600 and PS3-12 (38.18% and 38.07%, respectively). The genomic content encompasses 3614 protein-coding sequences, 86 transfer RNAs (tRNAs), and 9 ribosomal RNA (rRNA) operons with 6 different 16S rRNA sequences (Table 2).

Interestingly, the genome of strain Lsc_1132^T^ comprises 83.25% protein-coding genes, a proportion that is significantly higher than the 76.00% and 76.75% observed in strains EB600 and PS3-12, respectively. This suggests that strain Lsc_1132^T^ could have adapted to the nutrient-poor conditions of the deep-sea sediments by increasing the efficiency with which its genome encodes proteins to maximize the utilization of scarce resources in its environment.

Following the COG classification, a total of 2523 (69.81%) gene-encoding proteins were assigned to 22 categories (Appendix A). The predominant COG categories included amino acid transport and metabolism (COG-E, 9.05%), carbohydrate transport and metabolism (COG-G, 8.80%), translation, ribosomal structure, and biogenesis (COG-J, 7.42%), energy production and conversion (COG-C, 7.03%), and transcription (COG-K, 6.75%).

### 3.4. The Metabolic Characteristics of Strain Lsc_1132^T^

To investigate the metabolic characteristics and ecological functions of strain Lsc_1132^T^, we reconstructed the metabolic networks and conducted a comparative analysis with strains EB600 and PS3-12 from the same genus (Figure 5).

#### 3.4.1. The Utilization of Carbohydrates

In the streamlined genome of strain Lsc_1132^T^, most COG categories are present in lower numbers compared to the other two strains. However, the count of COG-Gs in strain Lsc_1132^T^ is significantly higher than in EB600, suggesting that the carbohydrate metabolism might play a crucial role in the survival of strain Lsc_1132^T^. In fact, strain Lsc_1132^T^ is equipped with as many as 87 genes in sugar transport, including GntP family gluconate:H+ symporter, ABC transport systems for multiple sugar, maltose (*musEFG*), ribose **(***rbsABC*), fructooligosaccharide (*fusABC*), inositol-phosphate (*inoEFGK*), raffinose/stachyose/melibiose (*msmEFG*), lactose/L-arabinose (*lacEFG*), methyl-galactoside (*mglABC*), arabinogalactan oligomer/maltooligosaccharide (*ganOPQ*), and sn-glycerol 3-phosphate (*ugpABCE*), as well as PTS systems for mannitol (*cmtAB*), glucitol/sorbitol (*srlAB*), galactitol (*gatAB*), cellobiose (*celABC*), and 2-O-A-mannosyl-D-glycerate (*mngAB*) (Appendix A).

Additionally, it possesses three extracellular glycoside hydrolase genes that belong to the GH3, GH13, and GH25 families, indicating that it could be capable of the extracellular breakdown and utilization of complex carbohydrates. We subsequently confirmed that arabinogalactan (Figure 6), as well as pectin, dextrin, D-maltose, D-trehalose, D-cellobiose, sucrose, D-turanose, stachyose, D-raffinose, *α*-D-lactose, D-melibiose, *β*-methyl-D-glucoside, D-salicin, *N*-acetyl-D-glucosamine, *N*-acetyl-*β*-D-mannosamine, *N*-acetyl-D-galactosamine, *N*-acetyl neuraminic acid, *β*-gentiobiose, *α*-D-glucose, D-mannose, D-fructose, D-galactose, 3-methyl glucose, D-fucose, L-fucose, L-rhamnose, and D-mannitol, can serve as the sole carbon source to support the growth of strain Lsc_1132^T^ (Appendix A). These pieces of evidence highlighted its adaptability to diverse sugar substrates in hadal ecosystems.

It is noteworthy that the transporter genes of sn-glycerol 3-phosphate, galactitol, inositol-phosphate, arabinogalactan oligomer/maltooligosaccharide, glucitol/sorbitol, raffinose, stachyose, melibiose, and methyl-galactoside were only found in strain Lsc_1132^T^ (Appendix A), but were absent in the other two strains. This suggests that the expansion of utilizable sugar types through horizontal gene transfer may be crucial for the survival of strain Lsc_1132^T^ in the hadal sediment environment.

#### 3.4.2. Metabolisms of Amino Acids

Regarding the biosynthesis of amino acids, strain Lsc_1132^T^ is equipped to biosynthesize the full spectrum of 20 amino acids. Specifically, strain Lsc_1132^T^ is endowed with a gene that encodes a bifunctional enzyme with both phosphoribosyl-AMP cyclohydrolase and phosphoribosyl-ATP pyrophosphohydrolase activities, known as *hisIE* (EC: 3.5.4.19 and 3.6.1.3). However, it lacks the genes involved in the histidine degradation pathway (*hutGHIU*). In contrast, strains EB600, PS3-12, as well as the outgroup strain PS3-40, exhibit an opposing genetic profile, possessing genes for histidine degradation but lacking the *hisI*, *hisE*, and *hisIE* genes. We further confirmed that strain Lsc_1132^T^ is capable of growth using the mixture of other 19 amino acids as both carbon and nitrogen sources. This observation suggests that the native environment of strain Lsc_1132^T^ might be deficient in histidine, requiring the strain to have its own synthesis capabilities.

As for the uptake and utilization of amino acids and their derivatives, strain Lsc_1132^T^ is predicted to have as many as 68 transporter genes, including ABC transport systems for polar amino acids, tryptophan/tyrosine, D-methionine, arginine/lysine/histidine, general branched-chain amino acids, glutamine, aspartate/glutamate/glutamine, oligopeptides, and dipeptides. In addition, it also harbors the gene-encoding APA family basic amino acid/polyamine antiporter, aromatic amino acid transporter (*aroP*), sodium/proline symporter (*putP*), lysine-specific permease (*lysP*), AGCS family alanine/glycine: cation symporter, AAT family amino acid permease, and proton/glutamate symporter (*gltP*) (Appendix A). We further confirmed that it could utilize L-aspartic acid, L-glutamic acid, L-pyroglutamic acid, and L-serine (Appendix A), and could grow with peptone as the sole carbon source. This implies that strain Lsc_1132^T^ could have a sophisticated system for acquiring and utilizing amino acids and their derivatives from its environment.

In addition to the L-type amino acids, strain Lsc_1132^T^ features two genes that encode transporters for D-serine/D-alanine/glycine (Appendix A). Its capability in D-serine utilization was further confirmed (Appendix A). Moreover, the strain possesses four genes of extracellular D-alanyl-D-alanine carboxypeptidases belonging to the S11 and M15 peptidase families (Appendix A). In addition, there are genes identified for peptidoglycan DD-metalloendopeptidase, murein DD-endopeptidase, and the cell wall hydrolase, known as SleB. These molecular inventories strongly suggest that strain Lsc_1132^T^ might be equipped to acquire D-amino acids through the enzymatic breakdown of cell wall components, which underscored a potential ecological niche for strain Lsc_1132^T^ in the decomposition of recalcitrant organic matter within the nutrient-depleted ecosystems of the hadal zone.

#### 3.4.3. Metabolisms of Organic Acids

We identified a total of ten carboxylate transport-associated proteins within the genome of strain Lsc_1132^T^, encompassing the C4-dicarboxylate ABC transporter system (*dctMPQ*), formate transporter (*fdhC*), cationic/acetate symporter (*actP*), and lactate permease (*lctP*) (Appendix A). Furthermore, strain Lsc_1132^T^ also harbors the genes encoding acetate kinase (ACF5W4_13970), lactate/malate dehydrogenase (ACF5W4_01225), and butyrate kinase (ACF5W4_12235), as well as a cluster coding formate dehydrogenase subunits (from ACF5W4_18090 to ACF5W4_18110).We further confirmed that lactate, D- and L-malic acid, *γ*-amino-butyric acid, *β*-hydroxy-D, L-butyric acid, *α*-keto butyric acid, acetoacetic acid, and quinic acid could be utilized by strain Lsc_1132^T^ (Appendix A). These findings collectively suggested its ecological function in the assimilation of various carboxylates in hadal ecosystems.

#### 3.4.4. Nitrogen Metabolism

Regarding its capacity for inorganic nitrogen assimilation, strain Lsc_1132^T^ is equipped with the genes of ammonia transporters and NarK family nitrate/nitrite transporters. It also harbors genes associated with the assimilatory nitrate reductase pathway, specifically the *nasABCDE* operon (Appendix A). Moreover, strain Lsc_1132^T^ contains genes for the membrane-bound nitrate reductase enzyme complex (*narGHI*), suggesting its capability in anaerobic respiration in anoxic sediments, although it lacks the genes for nitrite reductase and other downstream pathways. In addition, the presence of a gene-encoding polyamine transporter, ubiquitous nucleoside ABC transport system, NCS1 family nucleobase: cation symporter, cytosine permease, and NCS2 family adenine/guanine/hypoxanthine permease underscores its ability to assimilate a broad spectrum of organic nitrogenous compounds (Appendix A). Collectively, these features illustrated the metabolic versatility of strain Lsc_1132^T^ in nitrogen utilization, potentially aiding its adaptation to the oligotrophic conditions of hadal sediments.

## 4. Discussion


*Genomic Streamlining*
*of Strain Lsc_1132^T^*


Compared to the species in the same genus or even the neighboring genera, the smaller genomic size of strain Lsc_1132^T^ could be a significant characteristic of it. To investigate the genomic streamlining mechanisms of strain Lsc_1132^T^, we first analyzed the orthologous gene families of strains Lsc_1132^T^, PS3-12, and EB600, as well as the outgroup strain PS3-40. It revealed that strain Lsc_1132^T^ contains 3615 orthologous families, which are significantly fewer than the 4947 and 4889 found in strains EB600 and PS3-12, respectively. Moreover, strain EB600 and PS3-12 share 2373 and 2395 orthologous families with outgroup strain PS3-40, respectively, but merely 2192 and 2147 with strain Lsc_1132^T^, respectively (Appendix A). This suggests that strain Lsc_1132^T^ has undergone a significant loss of orthologous genes. This phenomenon is similar to the report that *Shewanella*, *Psychromonas*, and *Colwellia* in the hadal zone have smaller genomic sizes compared to those in the surface and mid-depths of the ocean [50]. This suggests that genome streamlining may be one of the common strategies for microorganisms to adapt to the extreme environment of the hadal zone.

To understand the evolutionary history of strain Lsc_1132, the gained and lost gene families were calculated for each phylogenetic node and branch of strains Lsc_1132^T^, EB600, PS3-12, and PS3-40, as well as 20 *Neobacillus* species (Appendix A). This showed that their most recent common ancestor (MCRA, Node 0) contains 5641 gene families, which experienced a gene acquisition event with 145 gene families to form the MCRA of genera “*Aliineobacillus*” and “*JAUZPL01*” (Node 1). Subsequently, Node 1 experienced a minor gain of 41 genes and a loss of 521 genes, resulting in the formation of the MRCA of the genus “*Aliineobacillus*” (Node 2). After gaining 680 gene families and losing a significant 2444 families, strain Lsc_1132 was ultimately formed.

According to the COG classification, during the formation of the MRCA of the genus “*Aliineobacillus*” (from Node 1 to Node 2), there was a significant net loss in the number of genes in the COG-E (energy production and conversion), COG-R (general function prediction only), COG-G (carbohydrate transport and metabolism), COG-K (transcription), and COG-M(cell wall/membrane/envelope biogenesis) categories, with losses of 51, 38, 29, 29, and 29 genes, respectively. Furthermore, during the speciation of Lsc_1132 (from Node 2 to strain Lsc_1132), the most significant net reduction in genes was observed in COG-R (general function prediction only), COG-E (amino acid transport and metabolism), and COG-F (nucleotide transport and metabolism), with losses of 160, 152, and 147 genes, respectively. Interestingly, during the speciation process of strain Lsc_1132, there was a net gain in the number of COG-X (Mobilome: prophages, transposons) genes, indicating that horizontal gene transfer played a significant role in the evolution of Lsc_1132 (Figure 7).

To understand the link between the compact genome of strain Lsc_1132^T^ and its adaptation to the deep-sea environment, we conducted a comparative metabolic analysis of the three strains based on KEGG annotations. The results indicate that there are 230 KEGG Orthology (KO) numbers that are shared by EB600 and PS3-12 but which are absent in strain Lsc_1132^T^. For example, strain Lsc_1132^T^ specially lacks the cytochrome c synthesis gene cluster, including *ccmBCEFGHZ*, indicating the absence of cytochrome c in the respiratory chain (Appendix A). This may be due to the inability of hadal sedimentary environment to provide sufficient oxygen to maintain a cytochrome c-dependent respiratory chain, and instead, it may utilize the menaquinone-shuttle systems to respire more efficiently with lower oxygen concentrations.

Although it was reported that many deep-sea bacteria can utilize aromatic compounds and other recalcitrant organic matter to cope with oligotrophic conditions, we identified a strain specific loss of 26 genes involved in the degradation of aromatic compounds in strain Lsc_1132^T^ (Appendix A). This gene loss encompasses genes that encode for benzoate transporters (*benK* and *benE*), along with the enzymes involved in the oxidation and degradation of various aromatic substrates, including catechol, gentisate, pcumate, phenylacetic acid, phenol, 3-hydroxybenzoate, benzene, and toluene. We propose that the low oxygen concentration in hadal sediments may impede the aerobic degradation of aromatic compounds by oxygenases, leading to the loss of these genes.

Additionally, strain Lsc_1132^T^ specifically lacks the genes for poly-beta-hydroxybutyrate (PHB) synthesis (*phaC* and *phbB*), which is a biopolymer used by some bacteria as an energy storage material (Appendix A). Considering the strain’s capacity for starch and glycogen synthesis, as well as its repertoire of many sugar transporters, we propose that employing starch or glycogen as carbon storage materials may be more energy efficient than PHB, which could be crucial for its survival in the oligotrophic environment of the hadal zone.

Besides orthologous families, we also examined the number of paralogous genes, which result from gene duplication events. We found that strain Lsc_1132^T^ possesses only 61 gene families with paralogs, significantly fewer than those of strains EB600 and PS3-12 (368 and 233, respectively). This suggests that genomic de-redundancy could be another factor contributing to the streamlining of strain Lsc_1132^T^’s genome, potentially reducing the costs associated with DNA replication without compromising metabolic function.

## 5. Conclusions

The novel *Bacillota* strain Lsc_1132^T^, recently isolated from the Mariana Trench at a depth of 10,954 m, represents a novel species of a novel genus, proposed as *Aliineobacillus hadale* gen. nov. sp. nov. (type strain Lsc_1132^T^ = MCCC 1K09620^T^). Metabolically, strain Lsc_1132^T^ exhibits a robust carbohydrate metabolism supported by a unique suite of sugar transporters. It also acquires deep-sea nutrient with a comprehensive transport system for amino acids and peptides, including a strain-specific histidine synthesis ability and an absence of histidine degradation pathways. Genomically, strain Lsc_1132^T^ has undergone a substantial loss of genes, resulting in a streamlined genome optimized for deep-sea adaptation. It notably lacks genes for cytochrome c synthesis, aromatic compound degradation, and poly-beta-hydroxybutyrate (PHB) synthesis, implying an adaptation to low oxygen and oligotrophic conditions through alternative metabolic pathways. Additionally, the lower number of paralogous genes in strain Lsc_1132^T^, combined with its high density in protein-coding genes, may facilitate further streamlining of its genome and potentially minimizing the energetic costs associated with DNA replication. This study represents a notable contribution to the catalog of hadal microorganisms, illustrating the intricate ways in which life can adapt and persist in the extreme conditions of the deep ocean.

### 5.1. Description of Aliineobacillus gen. nov.

*Aliineobacillus* (A.li.i.ne.o.ba.cil’lus: L. masc. pron. *alius*, other, different; Gr. masc. adj. *neos*, new; L. masc. n. *bacillus*, a small rod and *Bacillus*, a bacterial genus; N.L. masc. n. *Aliineobacillus*, a different and new *Bacillus*).

Cells were Gram-stain-positive, short rods, lacking flagella. The major cellular fatty acids were iso-C_15:0_, Summed Feature 3 (C_16:1_*ω*6c and/or C_16:1_ *ω*7c), iso-C_17:0_, anteiso-C_15:0_ and iso-C_17:1_ *ω*5c. The genomic DNA G + C content was 40.9%.

The type species is *Aliineobacillus hadale*.

### 5.2. Description of Aliineobacillus hadale sp. nov.

*Aliineobacillus hadale* (ha.da′le. N.L. neut. Adj. hadale from Greek Háidēs, hadal of or relating to the deepest regions of the ocean).

Cells were Gram-stain-positive, short rods (0.8–1.2 μm in width and 2–4 μm in length). These cells did not possess flagella. Growth was observed at temperatures between 15 °C and 45 °C (optimum 40 °C), at NaCl concentrations from 0 to 3% (optimum 0%), and at pHs from 5 to 8 (optimum 6). The major cellular fatty acids of strain Lsc_1132^T^ included iso-C_15:0_, Summed Feature 3 (C_16:1_*ω*6c and/or C_16:1_*ω*7c;), iso-C_17:0_, anteiso-C_15:0_, and iso-C_17:1_*ω*5c. The genomic DNA G + C content was 40.9%.

The type strain, Lsc_1132^T^ (=MCCC 1K09620^T^), was isolated from a hadal sediment sample collected at a depth of 10,954 m from the Mariana Trench (11.327° N; 142.188° E, site MBR02).

## Figures and Tables

**Figure 1 microorganisms-13-00132-f001:**
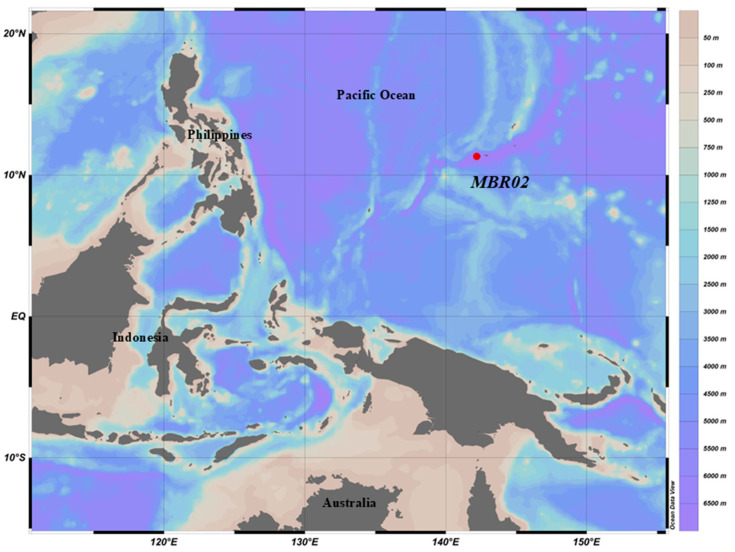
Map of the sampling site.

**Figure 2 microorganisms-13-00132-f002:**
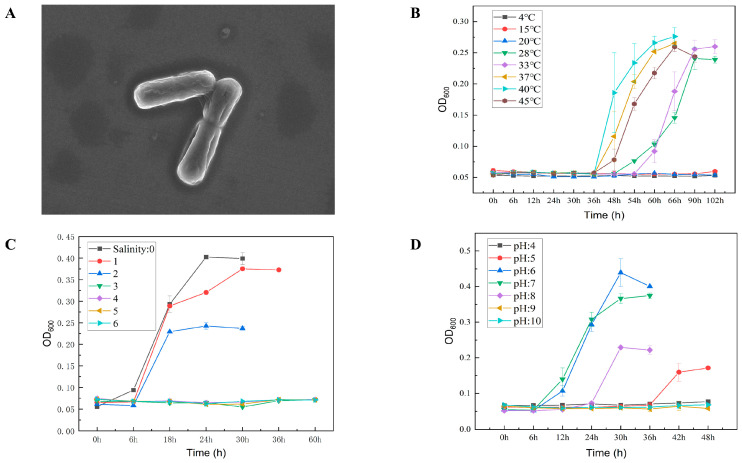
The transmission electron micrograph of strain Lsc_1132^T^ (scale bar: 5 µm) (**A**) and the growth curve of strain Lsc_1132^T^ at different temperatures (**B**), at different NaCl concentrations (**C**), and at different pH (**D**).

**Figure 3 microorganisms-13-00132-f003:**
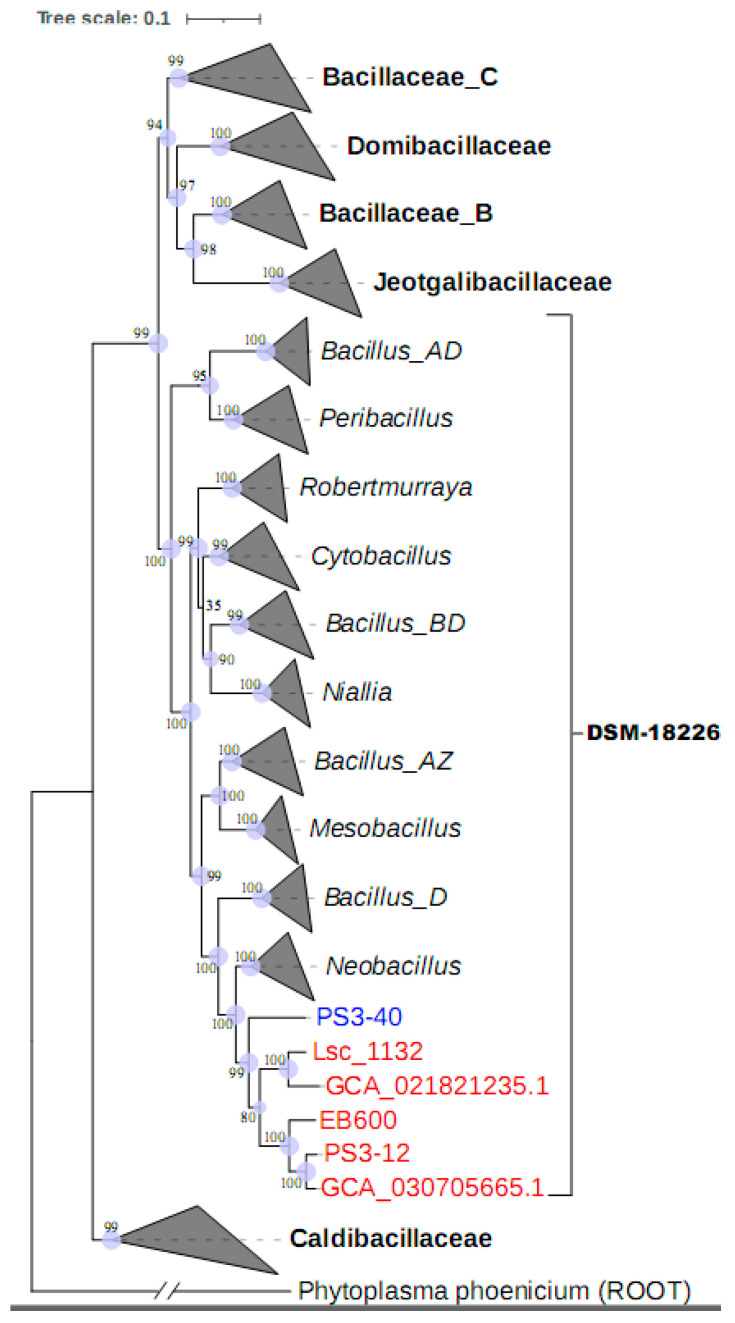
Phylogeny of the species and their related species based on 120 concentrated proteins. All families within GTDB-proposed order “*Bacillales*_B” are included. The genus “*JAEVLT01*” (proposed as “*Aliineobacillus*”) and “*JAUZPL01*” are highlighted in red and blue, respectively. The bootstrap values are also shown. The units in the tree scale bar represent amino acid substitutions per site.

**Figure 4 microorganisms-13-00132-f004:**
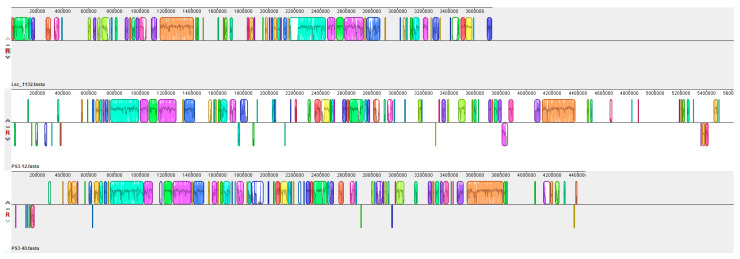
Genomic collinearity of strains Lsc_1132^T^, PS3-12 and PS3-40.

**Figure 5 microorganisms-13-00132-f005:**
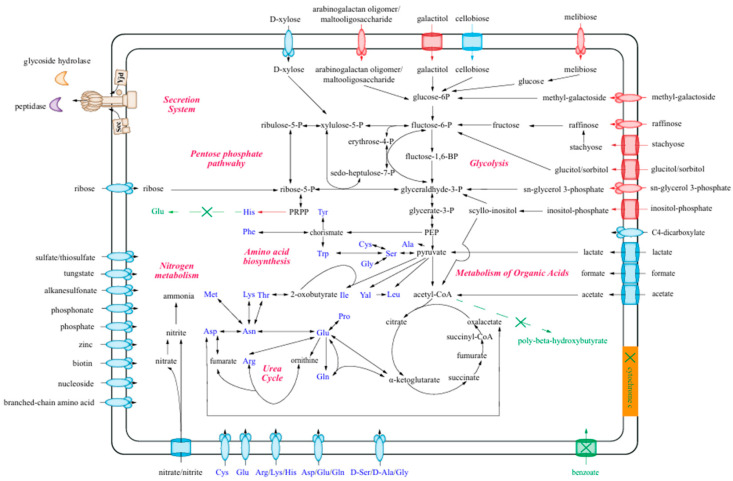
Reconstructed metabolic pathways of strain Lsc_1132^T^. The strain Lsc_1132^T^ specific pathways and transporters are highlighted in red, while the strain-specific missing pathways in strain Lsc_1132^T^ are highlighted in green.

**Figure 6 microorganisms-13-00132-f006:**
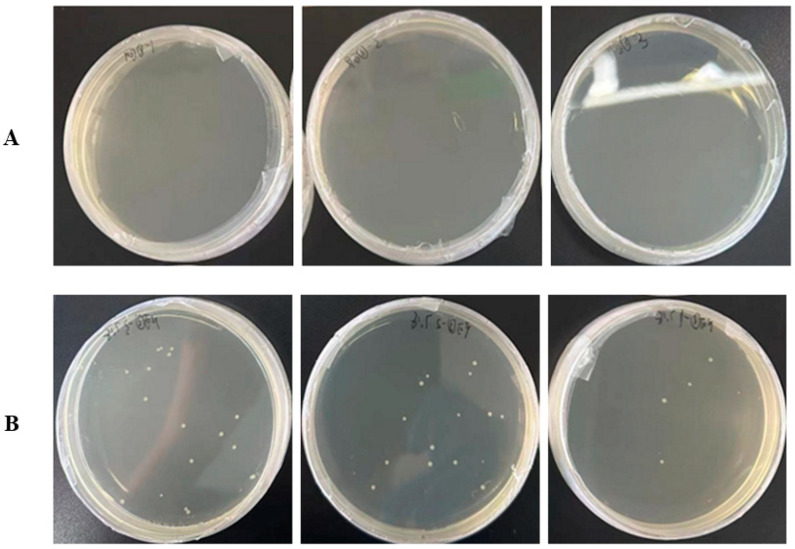
The arabinogalactan utilization of strain Lsc_1132^T^. (**A**) The subfigures represent LB agar plates coated with culture incubated for 0 days. (**B**) The subfigures represent samples with culture incubated for 5 days. Each sample was tested with three independent biological replicates.

**Figure 7 microorganisms-13-00132-f007:**
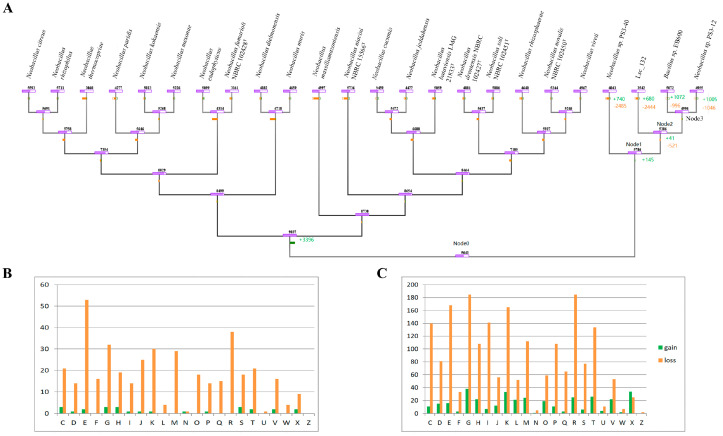
Dynamic evolution of gene families in strains Lsc_1132^T^, EB600, PS3-12, and PS3-40, as well as 20 *Neobacillus* species. (**A**) The numbers of gained and lost gene families for each node. (**B**,**C**) The identified COG functions of the gained and lost genes of the corresponding evolutionary events. Abbreviations: C, energy production and conversion; D, cell cycle control, cell division, chromosome partitioning; E, amino acid transport and metabolism; F, nucleotide transport and metabolism; G, carbohydrate transport and metabolism; H, coenzyme transport and metabolism; I, lipid transport and metabolism; J, translation, ribosomal structure, and biogenesis; K, transcription; L, replication, recombination, and repair; M, cell wall/membrane/envelope biogenesis; N, cell motility; O, posttranslational modification, protein turnover, chaperones; P, inorganic ion transport and metabolism; Q, secondary metabolites biosynthesis, transport, and catabolism; R, general function prediction only; S, function unknown; T, signal transduction mechanisms; U, intracellular trafficking, secretion, and vesicular transport; V, defense mechanisms; W, extracellular structures; X, mobilome: prophages, transposons; Z, cytoskeleton.

**Table 1 microorganisms-13-00132-t001:** Differential characteristics between strain Lsc_1132^T^ and five related strains.

Characteristic	1	2	3	4	5	6
Ranges (optimum) for growth:						
Temperature(°C)	15–45	10–45	16–50	10–50	0–50	16–50
NaCl (*w*/*v*, %)	0–3	0–10	0–5	0–15	0–15	0–5
pH	5–8	6.0–9.5	6.0–10.0	6.0–12.0	6.0–12.0	6.0–10.0
Fatty acids:						
anteiso-C_15:0_	5.5	33.5	21.8	12.8	12.5	20.5
iso-C_17:1_*ω*5ϲ	5.5	−	−	−	−	−
iso-C_17:0_	6.4	2.6	2.6	1.5	2.7	1.4
C_16:1_*ω*7c alcohol	0.3	2.4	3.1	5.0	3.4	2.3
anteiso-C_16:0_	0.2	−	−	−	−	−
anteiso-C_17:1_ A	1.9	−	−	−	−	−
iso-C_13:0_	4.3	0.2	−	−	−	−
Acid production from:						
glycerol	w	−	w	ND	ND	−
D-cellobiose	w	−	+	ND	ND	−
D-turanose	+	−	+	ND	ND	v
D-mannitol	+	−	+	ND	ND	−
L-fucose	w	v	−	ND	ND	−
DNA G + C content (mol%):	40.90	40.10	40.10	40.80	41.10	39.40

1, Strain Lsc_1132^T^; 2, *Neobacillus soli NBRC* 102451^T^ [20]; 3, *Neobacillus bataviensis* LMG 21833^T^ [20]; 4, *Neobacillus rhizophilus* FJAT-49825^T^ [21]; 5, *Neobacillus citreus* FJAT-50051^T^ [21]; 6, *Neobacillus drentensis* LMG 21831^T^ [20].  +, Positive; −, negative; w, weak; v, results vary between strains; ND, no data available.

**Table 2 microorganisms-13-00132-t002:** Genome features of strain Lsc_1132^T^.

Items	Description
Size (bp)	3,735,591
G + C content (%)	40.9
Genes assigned to COG	3075
Protein-coding genes	3614
16S rRNA	9
23S rRNA	9
5S rRNA	9
rRNA operons	9
tRNA genes	86
Gene islands	12

## Data Availability

The GenBank accession number for the complete genome sequence of strain Lsc_1132^T^ is CP172536.1.

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
