# Peer review of "Comparative Genomics Reveals Evidence of the Genome Reduction and Metabolic Potentials of *Aliineobacillus hadale* Isolated from Challenger Deep Sediment of the Mariana Trench"

_microorganisms, 2025, doi:10.3390/microorganisms13010132_

Round 1
Reviewer 1 Report
Comments and Suggestions for Authors
The manuscript titled "Comparative Genomics Reveals Evidence of Genome Reduction and Metabolic Potentials of a Novel Strain Within Bacillota Isolated from Challenger Deep Sediment of the Mariana Trench" represents a valuable contribution to the understanding of microbial adaptations in extreme environments. The study includes robust genomic, phylogenetic, and biochemical analyses, supporting the proposal of Aliineobacillus hadale gen. nov. sp. nov..
The methodology is well-detailed, and the findings are substantiated by comprehensive datasets, including genome sequencing and biochemical assays. However, several aspects require improvement to enhance the manuscript's clarity, methodological rigour, and contextual alignment with existing literature:
1. The discussion of microbial adaptation to the hadal zone is engaging, particularly the focus on genome streamlining and metabolic versatility. This could be further enriched by citing additional studies on microbial adaptations to low-oxygen and oligotrophic environments, such as those involving chemosynthesis-based survival strategies.
2. Please include a phylogenetic tree based on 16S rRNA sequences constructed using multiple methods (neighbor-joining, maximum parsimony, and maximum likelihood) in the supplementary materials for comprehensive phylogenetic analysis.
3. Include Average Amino Acid Identity (AAI) values comparing the novel strain with closely related strains to strengthen taxonomic placement.
4. The growth range (10–45°C, with an optimum at 37°C) and the ability to grow without NaCl are unusual for a deep-sea isolate. Please provide a detailed explanation of these observations and discuss their implications for ecological adaptation.
5. Did you assess the strain's growth under anaerobic conditions? If not, consider performing these experiments or clarifying the aerobic growth conditions in the manuscript.
6. In Figure 2, improve the contrast and labeling in the growth curves to make the data easier to interpret.
7. Highlight transporter genes unique to Lsc_1132 in a visually distinct manner (e.g., using bold labels or a distinct color scheme in tables/figures) to emphasize their ecological significance.
8. The results and discussion sections occasionally overlap, making it challenging to differentiate findings from interpretations. For example, consider separating genome streamlining results from the discussion of their evolutionary implications to improve clarity.
9. It would be better to include the proposed novel bacterial name in the title to ensure clarity and alignment with the manuscript's focus.
10. Provide two culture deposit numbers from culture collections in two different countries, as required by the International Code of Nomenclature of Prokaryotes (ICNP).
11. In the introduction, include details about members of the Bacillaceae family, particularly Neobacillus, that have been reported from deep-sea or marine environments. Discuss their adaptability to extreme conditions to better contextualize and relate these findings to your novel isolate.
Author Response
Reviewer 1
The manuscript titled "Comparative Genomics Reveals Evidence of Genome Reduction and Metabolic Potentials of a Novel Strain Within Bacillota Isolated from Challenger Deep Sediment of the Mariana Trench" represents a valuable contribution to the understanding of microbial adaptations in extreme environments. The study includes robust genomic, phylogenetic, and biochemical analyses, supporting the proposal of Aliineobacillus hadale gen. nov. sp. nov.
The methodology is well-detailed, and the findings are substantiated by comprehensive datasets, including genome sequencing and biochemical assays. However, several aspects require improvement to enhance the manuscript's clarity, methodological rigour, and contextual alignment with existing literature:
1.The discussion of microbial adaptation to the hadal zone is engaging, particularly the focus on genome streamlining and metabolic versatility. This could be further enriched by citing additional studies on microbial adaptations to low-oxygen and oligotrophic environments, such as those involving chemosynthesis-based survival strategies.
Thanks for your great comments. We added “This phenomenon is similar to the report that Shewanella, Psychromonas, and Colwellia in the hadal zone have more streamlined genomes compared to those in the surface and mid-depths of the ocean. This suggests that genome streamlining may be one of the common strategies for microorganisms to adapt to the extreme environment of the abyss” from line 477 to 481.
2.Please include a phylogenetic tree based on 16S rRNA sequences constructed using multiple methods (neighbor-joining, maximum parsimony, and maximum likelihood) in the supplementary materials for comprehensive phylogenetic analysis.
Changed as you suggested. The results are presented in Supplementary Figure S1.
3.Include Average Amino Acid Identity (AAI) values comparing the novel strain with closely related strains to strengthen taxonomic placement.
Changed as you suggested. AAI are described in line 294-295 and Supplementary Table S6.
4.The growth range (10–45°C, with an optimum at 37°C) and the ability to grow without NaCl are unusual for a deep-sea isolate. Please provide a detailed explanation of these observations and discuss their implications for ecological adaptation.
Thanks for your great comments. If most ocean scientists found that it is a common phenomenon that there are many paradoxes, for example, optimum growth pressure, optimum growth temperature and so on. It is difficult to know where these microbes come from deep-sea sediment in-situ or falling from the surface water column? Therefore, the strain Lsc 1132 had SpoA gene that can produce Spore to adapt ecological environments. Another explanation is that some strains from trenches grow too slowly, therefore, generation time maybe need several months or years. Thanks for your consideration again.
- Did you assess the strain's growth under anaerobic conditions? If not, consider performing these experiments or clarifying the aerobic growth conditions in the manuscript.
Thanks for your great comments. The strain can carry out dissimilatory iron reduction using glucose under anaerobic conditions in our unpublished data.
- In Figure 2, improve the contrast and labeling in the growth curves to make the data easier to interpret.
Changed as you suggested, we updated the Figure 2.
7.Highlight transporter genes unique to Lsc_1132 in a visually distinct manner (e.g., using bold labels or a distinct color scheme in tables/figures) to emphasize their ecological significance.
Thanks for your comments. Acturally, we have highlighted the strain-specific transporters in the figure about metabolism and the figure legend was modified at the same time.
8.The results and discussion sections occasionally overlap, making it challenging to differentiate findings from interpretations. For example, consider separating genome streamlining results from the discussion of their evolutionary implications to improve clarity.
Thanks for your great comments. We moved some results of genome streamlining into discussion. Moreover, we further added some description about the gain and loss events of genes during evolution of its related species.
- It would be better to include the proposed novel bacterial name in the title to ensure clarity and alignment with the manuscript's focus.
Thanks for your comments, we modified the title as ‘Comparative Genomics Reveals Evidence of Genome Reduction and Metabolic Potentials of Aliineobacillus Hadale Isolated from Challenger Deep Sediment of the Mariana Trench’.
- Provide two culture deposit numbers from culture collections in two different countries, as required by the International Code of Nomenclature of Prokaryotes (ICNP).
The strain Lsc_1132 has been sent to the Korean Culture Center of Microorganisms (KCTC). Once we receive the accession number from KCTC, the data will be included in the manuscript.
- In the introduction, include details about members of the Bacillaceae family, particularly Neobacillus, that have been reported from deep-sea or marine environments. Discuss their adaptability to extreme conditions to better contextualize and relate these findings to your novel isolate.
Thanks for your great comments. Changed as you suggested and we added two paragraphs in line 81-95.

Reviewer 2 Report
Comments and Suggestions for Authors
The article is interesting. Important issues should be addressed, for the methodological and results sections, to refine the work.
Major comments
1) It is important to include the sequence identity values of the complete 16S rRNA ribosomal gene for this type of study in which new taxonomic categories are proposed. Partial gene sequence identity values are presented in the manuscript. Values could also be obtained and indicated from the search for this marker in the recovered genome. Maybe I was a bit confused, but was this variation in the sequence of the 16S rRNA ribosomal gene also detected in the assembled genome? I suggest in any case to improve the explanation of this phenomenon and its detection.
2) It is considered relevant to include figures of genome alignments between the new strain with the closest strains that are sequenced, as well as with the type strains of nearby species. This may help to identify in more detail the evolution of the genomes, especially the conserved syntenic blocks and the regions that may have been lost in the narrower genome. This could include alignments generated with the Mauve tool, for example, which is common, or with others.
3) It would be highly recommended to include references to articles in which taxonomically close species have been described to refer to the cut-off points for species and genus definition using the 16S rRNA ribosomal gene molecular marker. The most recognized cut-off point for genus level is between 93-94% identity in the complete ribosomal gene between type species, but this may change in some bacterial species.
4) In the introduction it is mentioned that this group of bacteria can form spores and have genes related to resistance to heavy metals. Does the recovered genome have the genetic potential to carry out these functions? As this is indicated in the introduction, as a characteristic of the group, and is of interest because of the type of extreme environment, one questions this when arriving at the results.
5) It would be interesting to know what other bacterial strains could be identified during the culture experiments. Was there only one strain related to strain Lsc_1132? What was the motivation for studying particularly this strain and not others? Were several related colonies identified or was it only one colony that was studied?
6) In the introduction it is mentioned that extensive analyses were performed to characterize the phylogenetic position of the strain studied. However, only one analysis is presented in the methods. I suggest shortening the sentences or including the other phylogenetic analyses. Especially for this type of taxonomic studies, and considering that new taxonomic categories are proposed, it is important to include a traditional phylogenetic tree using complete 16S rRNA ribosomal genes, including type strain sequences for species and genera, which may be richer in sequences than only those that could be included with complete genomes. This tree should be performed with a robust methodology that determines the best surrogate model and bootstrap replicates. Other methodologies could be used to confirm strain position, such as PhyloPhlan, which is easy to use, or other complementary phylogenomic methods.
7) In the methods, on the phylogenetic analysis it is mentioned that a manual curation of the food was performed. The alignment is not deposited in a public repository, so it is not possible to replicate this part of the methodology. In general, the use of manual curation of the sequences is not recommended; it is recommended to use other types of strategies or tools, for example, the use of GBlocks or similar programs that allow a standard and replicable curation of the alignments.
8) Can you explain a little more about these names: -labeled as genus “JAEVLT01” within the family “DSM-18226”- (lines 258-259). What do they refer to? do they have any published reference? where do they come from? are they taxonomically valid names?
9) It is important to include the supporting values in the phylogenetic tree (Figure 3). I do not consider that it is rigorous to group all the nodes in a category greater than 80 values, especially in the branches that go towards the strains of interest. The values should be visible. In some studies, even branch distances are used to support taxonomic proposals. This also applies to the name "JAUZPL01” in line 279.
10) The manuscript does not provide information on the number of readings obtained for either Illumina or PacBio. The methods used for sequence curation by quality are not indicated. The complete bioinformatics programs used to obtain the genome are not indicated. The coverage of the genome obtained is not indicated. These data are very important to indicate in this type of work.
11) Please include more informative figure legends, in general. In particular, for figure 4 it would be important to describe in detail the figure, for example what the inner circles represent, as well as detailing any outstanding feature that the genome has that can be seen with this figure. In my opinion, this figure can be supplementary material, and can be replaced by a Mauve genome alignment to see the changes that occur in the genomic order, the syntenic blocks, the missing areas of the genome that make it more compact, it would be more informative for the work.
12) The manuscript focuses on análises between Lsc_1132, E600, PS3-12 and PS3-40 strains, which is important. It would also be important to find the differences of these strains, with the others taxonomically closer to be able to define in more detail the differences or characteristics of the genus.
Minor comments
1) It is important to identify sequences corresponding to type strains, in the manuscript and in the figures, with the superscript T.
2) I suggest including a sentence in the methods about the sample collection, that further reinforces the idea that the sample was not contaminated during the trip back to the surface. Perhaps explain a bit more about what the subsampling of the sample consisted of, was the less exposed “inside” of the sediment sampled?
3) Please correct “speicies” in line 260.
4) It is important to state the units in the tree scale bar. “nucleotide substitutions per site”, for example.
5) Please indicate in the phylogenetic tree where the new genus is proposed, perhaps put an index or a line indicating from where to where the proposed genus extends, or put the proposed names in parentheses.
6) Please consider the use of AAI metrics to enrich your manuscript.
Author Response
Reviewer 2
The article is interesting. Important issues should be addressed, for the methodological and results sections, to refine the work.
Major comments:
1) It is important to include the sequence identity values of the complete 16S rRNA ribosomal gene for this type of study in which new taxonomic categories are proposed. Partial gene sequence identity values are presented in the manuscript. Values could also be obtained and indicated from the search for this marker in the recovered genome.
Thanks for your comments. We added a new Supplementary Table S3 to describe the 16S RNA gene similarity. These 16S rRNA sequences all are predicted from the genomes and with length >=1500 nt. The detailed M&M are also included in line 208-213 and line 288-292.
Maybe I was a bit confused, but was this variation in the sequence of the 16S rRNA ribosomal gene also detected in the assembled genome? I suggest in any case to improve the explanation of this phenomenon and its detection.
Thanks for your comments. The heterogeneity of 16S rRNA gene sequences in microorganisms is very common. For example, we found that Shewanella psychropiezotolerans YLB-06 contains 12 16S rRNA gene sequences with 11 sequence variants. Also, Shewanella sp. MTB7 contains eleven 16S rRNA, none of which showed 100% identity with each other (ref: Genomic Analysis of the Deep-Sea Bacterium Shewanella sp. MTB7 Reveals Backgrounds Related to Its Deep-Sea Environment Adaptation).
This phenomenon is difficult to explain. At least, we proposed that it could be related to environmental resistance (another manuscript under review). Acinetobacter sp. A1-4-2 isolated from a crab farmming pool shows sequence uniformity in its 16S rRNA genes, while its closely related strain, Acinetobacter sp. CS-2, exhibits four unique variants of the 16S rRNA gene (please see the phylogenetic tree below to see how closely related they are). Actually, Acinetobacter sp. CS-2 was isolated from hospital wastewater and identified as a super antibiotic-resistant bacterium. We proposed that this genetic heterogeneity in strain CS-2 could conceivably enhance its defensive capabilities against antibiotics that interfere with ribosomal activity, such as aminoglycosides (e.g., gentamicin, streptomycin, neomycin) which bind to specific sites on the 16S rRNA, disrupting the process of bacterial protein synthesis, and tetracyclines (e.g., tetracycline, doxycycline) which bind to the 16S rRNA on the 30S ribosomal subunit, preventing the entry of aminoacyl-tRNA into the A site of the ribosome and thereby inhibiting protein synthesis.
Another example is about a bacterium in desert has two different 16S rRNA genes, one is transcribed in day time (hot), and the other is transcribe at night (cold).
2) It is considered relevant to include figures of genome alignments between the new strain with the closest strains that are sequenced, as well as with the type strains of nearby species. This may help to identify in more detail the evolution of the genomes, especially the conserved syntenic blocks and the regions that may have been lost in the narrower genome. This could include alignments generated with the Mauve tool, for example, which is common, or with others.
Thanks for your comments. These results are added in Figure 4.
3) It would be highly recommended to include references to articles in which taxonomically close species have been described to refer to the cut-off points for species and genus definition using the 16S rRNA ribosomal gene molecular marker. The most recognized cut-off point for genus level is between 93-94% identity in the complete ribosomal gene between type species, but this may change in some bacterial species.
Thank you for your thorough review and the valuable comments. In this study, strain Lsc_1132 was found to have highest sequence similarity to Neobacillus bataviensis LMG 21833T (AJ542508) (99.10%), N. dielmonensis (HG315676) (99.02%) and N. drentensis NBRC 102427T (AJ542506) (99.01%). These values are observely higher than genus-level cutoffs. However, we believe that using 16S sequences for taxonomy positioning can sometimes be very ambiguous. For example, we found that the 16S rRNA similarity between “Thalassolituus sp. TMPB967” and the Thalassolituus type strain T. oleivorans MIL-1T was 95.74%. But based on GTDB taxomony, we classified TMPB967 into Venatorbacter but not Thalassolituus (ref: The phylogeny and metabolic potentials of an n-alkane-degrading Venatorbacter bacterium isolated from deep-sea sediment of the Mariana Trench). Considering that we have obtained the complete genome sequences of strain Lsc_1132 to make a genome tree, we do not plan to add such cut-off points prove something.
4) In the introduction it is mentioned that this group of bacteria can form spores and have genes related to resistance to heavy metals. Does the recovered genome have the genetic potential to carry out these functions? As this is indicated in the introduction, as a characteristic of the group, and is of interest because of the type of extreme environment, one questions this when arriving at the results.
Thanks for your comments. Some species of genus bacillus had versatile functional in marine sediments, such as iron reduction, removing heavy metals et al. In this study, we just focus on genome function. The genome of strain Lsc_1132 had SpoA gene that can produce spores. Others related studies will be carried out in the future.
5) It would be interesting to know what other bacterial strains could be identified during the culture experiments. Was there only one strain related to strain Lsc_1132? What was the motivation for studying particularly this strain and not others? Were several related colonies identified or was it only one colony that was studied?
Thanks for your comments. We completed National Key R&D Program Project "Deep Sea Key Technology and Equipment" and obtained more than 4000 strains from different trenches in the world. In this study, we focused on strain Lsc_1132 because it had obvious characters. Phylogenetically, the strain represents a novel genus, which is also helpful to reclassify “Neobacillus” sp. PS-12 and “Bacillus” sp. EB600. Moreover, its genomic streamlining is very interesting, which can reduce the energy cost during DNA duplication and could be strategy to survive in hadal environment. In addition, our previous research indicates that during the evolution of microorganisms from the surface ocean to the hadal zone, there is a trend of genome size initially increasing and then decreasing (ref: Phylogenomic analysis reveals a two-stage process of the evolutionary transition of Shewanella from the upper ocean to the hadal zone). However, abyssal whole-genome data is very limited. The streamlining of Lsc_1132 may further prove our hypothesis.
6) In the introduction it is mentioned that extensive analyses were performed to characterize the phylogenetic position of the strain studied. However, only one analysis is presented in the methods. I suggest shortening the sentences or including the other phylogenetic analyses. Especially for this type of taxonomic studies, and considering that new taxonomic categories are proposed, it is important to include a traditional phylogenetic tree using complete 16S rRNA ribosomal genes, including type strain sequences for species and genera, which may be richer in sequences than only those that could be included with complete genomes. This tree should be performed with a robust methodology that determines the best surrogate model and bootstrap replicates. Other methodologies could be used to confirm strain position, such as PhyloPhlan, which is easy to use, or other complementary phylogenomic methods.
Thanks for your comments. The results are added in the Supplementary Figure S2.
7) In the methods, on the phylogenetic analysis it is mentioned that a manual curation of the food was performed. The alignment is not deposited in a public repository, so it is not possible to replicate this part of the methodology. In general, the use of manual curation of the sequences is not recommended; it is recommended to use other types of strategies or tools, for example, the use of GBlocks or similar programs that allow a standard and replicable curation of the alignments.
Changed as you suggested. We reconstructed a phylogenetic tree that we eliminated positions with gap percentages of 50% or higher with trimAL.
8) Can you explain a little more about these names: -labeled as genus “JAEVLT01” within the family “DSM-18226”- (lines 258-259). What do they refer to? do they have any published reference? where do they come from? are they taxonomically valid names?
Thank you for your thorough review and the valuable comments. This branch was closely related to but not included in Neobacillus, which currently lacks a taxonomically valid name and is designated as genus "JAEVLT01" within the family "DSM-18226" of the order "Bacillales_B" according to the GTDB taxonomy.
9) It is important to include the supporting values in the phylogenetic tree (Figure 3). I do not consider that it is rigorous to group all the nodes in a category greater than 80 values, especially in the branches that go towards the strains of interest. The values should be visible. In some studies, even branch distances are used to support taxonomic proposals.
Thanks for your comments. We reconstructed a phylogenetic tree with bootstrap values in figure 3.
This also applies to the name "JAUZPL01” in line 279.
Thanks for your comments. We modified it as “genus "JAEVLT01" that also lacks a taxonomically valid name and is closely related to strain Lsc_1132 (Figure 3)”.
10) The manuscript does not provide information on the number of readings obtained for either Illumina or PacBio. The methods used for sequence curation by quality are not indicated. The complete bioinformatics programs used to obtain the genome are not indicated. The coverage of the genome obtained is not indicated. These data are very important to indicate in this type of work.
Changed as you suggested. We added related methods in line 192-206.
11) Please include more informative figure legends, in general. In particular, for figure 4 it would be important to describe in detail the figure, for example what the inner circles represent, as well as detailing any outstanding feature that the genome has that can be seen with this figure. In my opinion, this figure can be supplementary material, and can be replaced by a Mauve genome alignment to see the changes that occur in the genomic order, the syntenic blocks, the missing areas of the genome that make it more compact, it would be more informative for the work.
Thanks for your comments. We added a mauve result in line 331-334 and Figure 4. Also, we moved the genome ciros figure into Supplementary Figure.
12) The manuscript focuses on análises between Lsc_1132, E600, PS3-12 and PS3-40 strains, which is important. It would also be important to find the differences of these strains, with the others taxonomically closer to be able to define in more detail the differences or characteristics of the genus.
Thanks for your professional suggestion. We added new analysis and a new figure to describe the gene gain and loss events during the evolution of Lsc_1132, which included 20 Neobacillus genomes as reference. Please see line 485-506. Also, the Venn figure was moved to Supplementary Figure.
- It is important to identify sequences corresponding to type strains, in the manuscript and in the figures, with the superscript T.
Thanks for your comments. We published more than 20 papers in IJSEM in the past. Corresponding author have rich experience about these type strains with the superscript T. In this study, we highlighted the ecology function and adaption with genome methods. Therefore, some NOT type strains were applied.
2) I suggest including a sentence in the methods about the sample collection, that further reinforces the idea that the sample was not contaminated during the trip back to the surface. Perhaps explain a bit more about what the subsampling of the sample consisted of, was the less exposed “inside” of the sediment sampled?
Thanks for your comments. In the past 20 years, some reviewers often dispute whether the samples were contaminated in sampling or subsampling process. Now, it is very matured skill to do that. Thanks for your consideration.
3) Please correct “speicies” in line 260.
Changed.
4) It is important to state the units in the tree scale bar. “nucleotide substitutions per site”, for example.
Thanks for your comments. We added it in the legend of Figure 3.
5) Please indicate in the phylogenetic tree where the new genus is proposed, perhaps put an index or a line indicating from where to where the proposed genus extends, or put the proposed names in parentheses.
Thanks for your comments. We added it in the legend of Figure 3.
6) Please consider the use of AAI metrics to enrich your manuscript.
Thanks for your comments. The AAI results were added in line 294-295 and Supplementary Table S5.

Round 2
Reviewer 1 Report
Comments and Suggestions for Authors
The revised manuscript is much improved, with well-organized and clear findings. The authors have effectively addressed previous concerns, enhancing the quality of the work. I recommend it for publication.
Reviewer 2 Report
Comments and Suggestions for Authors
Thank you for the answers and for attending to the comments. The manuscript was enriched by the additions you made, particularly for discussion. Very interesting research.
A minor detail, a W should be changed to lower case in this edited phrase: After alignment, We eliminated positions.
Best